# Use of the Secreted Proteome of *Trametes versicolor* for Controlling the Cereal Pathogen *Fusarium langsethiae*

**DOI:** 10.3390/ijms20174167

**Published:** 2019-08-26

**Authors:** Alessia Parroni, Agnese Bellabarba, Marzia Beccaccioli, Marzia Scarpari, Massimo Reverberi, Alessandro Infantino

**Affiliations:** 1Dipartimento di Biologia Ambientale, Sapienza Università di Roma, P.le Aldo Moro 5, 00185 Rome, Italy; 2CREA-DC, via C.G. Bertero 22, 00156 Roma, Italy

**Keywords:** *Fusarium langsethiae*, exo-proteome, *Trametes versicolor*, T2-HT2, mycotoxins, growth inhibition

## Abstract

*Fusarium langsethiae* is amongst the most recently discovered pathogens of small grains cereals. *F. langsethiae* is the main producer, in Europe, of T2 and HT-toxins in small grain cereals, albeit often asymptomatic; this makes its control challenging. The European Union (EU) is pushing hard on the use of biocontrol agents to minimize the use of fungicides and pesticides, which are detrimental to the environment and responsible for serious pollution of the soil and superficial water. In line with EU directives (e.g., 128/2009), here we report the use of protein fractions, purified from the culture filtrate of the basidiomycete *Trametes versicolor*, for controlling *F. langsethiae*. *T. versicolor*, a so-called medicinal mushroom which is applied as a co-adjuvant in oncology and other pathologies as a producer of biological response modifiers. In this study, the exo-proteome of *T. versicolor* proved highly efficient in inhibiting the growth of *F. langsethiae* and the biosynthesis of the T2 toxin. Results are promising for its future use as a sustainable product to control *F. langsethiae* infection in cereals under field conditions.

## 1. Introduction

Fungi are organisms comprising yeasts, molds and mushrooms and they have been used for long time both in medicine as co-therapy and as food for their nutritional value. Notably, traditional Chinese medicine considered mushrooms (in particular basidiomycetes) as a useful source of bioactive compounds of large interest in medicine and to strengthen the welfare of the human body [1]. Polysaccharides, proteins, glycoproteins and peptides from different mushrooms (i.e., *Lentinula edodes*, *Ganoderma lucidum*, *Trametes versicolor*) demonstrated antibacterial, antiviral, antitumor and immune modulatory activity. One of the most known representatives of these compounds is the lentinan, a cell wall polysaccharide extracted from *Lentinula edodes* known for its medicinal properties [2,3,4]. Moreover, recent studies demonstrate a promising preclinical antileukemia activity of Tramesan, a patented α-hetero-polysaccharide purified from culture filtrate of *T. versicolor* [5]. Other biological effects were also investigated on murine cell line of melanoma (B16) where Tramesan showed, due to its reactive oxygen species (ROS)—indirect—scavenging ability, a significant limitation of cell growth. Tramesan, in fact, increased melanin content enhancing *nrf*-2 expression and protecting melanocytes against the dangerous ROS effect (due to the high intrinsic oxidative stress expressed by cancer cells) and, finally, a significant reduction of cell growth [6]. Another group of molecules with medicinal properties, such as proteins and peptides from higher basidiomycetes, has attracted the interest of the scientific community. In particular proteases, defensins, lectins, laccases, polysaccharopeptides and immune modulatory proteins showed different medicinal properties [1]. A particular family of immune modulatory proteins similar to phytohemagglutinins and immunoglobulins from *Ganoderma lucidum* and other mushrooms are denominated FIP (fungal immunomodulatory proteins). They are small protein molecules with a molecular weight of about 13kD with immune regulating activity [7] and the number of proteins belonging to FIP family is continuously increasing.

Several studies, however, mainly deal with the bioactivity of mushroom compounds as therapeutic tools while few studies regard the control of plant diseases. Plant pathogens and mycotoxins are organisms and metabolites causing severe economic losses in agriculture; indeed, mycotoxins are dangerous for animals and humans, often showing toxic or carcinogenic effects towards various target organs (i.e., aflatoxins vs. liver, ochratoxins vs. kidney, trichothecenes vs. lymphoid organs). In previous studies, we focused our attention on non-toxic or edible basidiomycetes such as *Lentinula edodes* and *T. versicolor.* Polysaccharide fraction purified from culture filtrates of *L. edodes* showed a significant inhibiting effect on aflatoxin synthesis by the plant pathogens *Aspergillus flavus* and *A. parasiticus* [8,9,10] and the same inhibiting effect on aflatoxin synthesis was demonstrated by rough and partially purified extracts from *T. versicolor* [11,12]. Extracts from both mushrooms promoted antioxidant defenses of the fungal cells consequently inhibiting toxin synthesis. In other studies, Tramesan was effective in the control of the infection of wheat leaves by the pathogen *Parastagonospora nodorum* [6] by strengthening the defenses of plant against the fungal pathogen. Nevertheless, Tramesan was non-toxic for fungal pathogens, whereas using the whole filtrate (i.e., without any purification step) showed an interesting fungitoxic effect (M. Reverberi, personal communication). The promising results obtained with these studies prompted us to investigate the protein fractions (up to 15% of the total filtrate d.w.) of *T. versicolor* as source of bioactive compounds and their possible efficacy in controlling other fungal pathogens and toxins synthesis. The idea to study the exo-proteome was also reinforced by the biological role played by FIP proteins from mushrooms. Previous research, in fact, evidenced a significant efficacy of oxidase enzymes (i.e., laccases), from the culture filtrates of an isolate of *T. versicolor*, in the detoxification and degradation of different toxins such as aflatoxins, ochratoxin A, Fusarium toxins (Deoxynivalenol and Fumonisin B1) at different inhibiting levels [13].

*Fusarium langsethiae* is an ascomycete located in the *Gibberella*-clade of *Fusarium,* in which several pathogens of small grains cereals (such as oat, wheat and barley) are present, which are very detrimental for crop quality and safety [14]. These pathogens are widespread worldwide and are actually producers of different toxins; some of which are even regulated by EC to limit their dangerous effects on humans and animals [15,16]. *Fusarium langsethiae* can be included among the new species discovered and studied over recent years, being the main producer in Europe of the T2 and HT2 toxins (type A trichothecenes) [15,17] found on small grain cereals [18]. It is difficult to investigate the pathogenicity of this fungus due to lack of visible symptoms on the infected plants. For this behavior, it is considered as an endophyte and the detection of the fungal infection has to be implemented with molecular techniques. In Italy, the pathogen incidence is higher in Centre and Southern than in Northern regions being related to climate conditions characterized by high temperature and scarce rain fallings during the wheat flowering [16]. The toxins T2 and HT2 produced by *F. langsethiae* present high toxicity probably due to their lipophilicity and are consequently likely to penetrate the cells [19]. Their toxic effects regard the inhibition of synthesis of DNA and RNA and proteins and the reduction of lymphocytes and immune defenses [15]. Moreover, they can induce lipid peroxidation affecting the membrane functions of the cell infected [20]. T2 toxin causes the ATA (Alimentary Toxic Aleukia) in humans.

Therefore, contamination of small cereals such as wheat with *F. langsethiae* may represent a serious concern for human and animal health. Reducing the incidence of *F. langsethiae* in small grain cereals would represent an important goal to be achieved for the safe and security of foods and feeds. Scarce literature is available on the fight against *F. langsethiae.* This is probably due to the difficulty to detect pathogenicity symptoms. At the same time, scarce reports are available about bioagents efficient against *F. langsethiae.* These reasons prompted us to investigate bioagents from *T. versicolor* to control this pathogen. In this study, we report the ability of different fractions of *Trametes versicolor* exo-proteome to inhibit *F. langsethiae* growth and toxin synthesis; we here propose these fractions as a possible biocontrol tool for managing this pathogen and its toxins.

## 2. Results

### 2.1. Culture Filtrate (CF) of Trametes Versicolor Inhibits the Growth of Fusarium Langsethiae

The preliminary assays concerned the effect of cultural filtrate (CF) of *T. versicolor* added to PDA medium in Petri dishes at concentrations of 0.04% *w*/*v* and 0.08% *w*/*v* on the growth of the pathogen *F. langsethiae* incubated at 25 °C for 3, 5 and 7 days after incubation (dai) (Appendix A). The growth of the pathogen resulted in the inhibition of 53.8% at the concentration of 0.04% *w*/*v* and 61.4% at the concentration of 0.08% *w*/*v* after 5 dai in comparison with the untreated sample (Figure 1). At 7 dai, however, the pathogen growth slightly increased even if significant inhibition was maintained in comparison with the control (*t* test; *p* < 0.001 for both treatments versus control). The fungal growth was estimated by diameter growth.

### 2.2. SDS-PAGE of the Exo-Proteome of Trametes Versicolor

The encouraging results obtained on the pathogen growth inhibition pushed us to characterize the protein fraction of the CF TV117. In fact, we already know that the polysaccharide fraction of the CF did not show fungal growth inhibition [6,9,11]. To this aim, proteins present in CF TV117 were precipitated with ammonium sulfate (AS) at 75 and 90% concentrations. The 2 protein fractions (F90AS and F75AS) obtained were analyzed by SDS-PAGE (Figure 2). In general, F90AS presented a lower type and quantity of proteins in comparison with F75AS fraction. This was confirmed by Bradford assay (data not shown). The molecular weights of the fractions assayed are in the range of 40–75 kDa in the F90AS fraction—the higher quantity of proteins localized at about 50 kDa—whereas the proteins in F75AS were distributed in a wider range—from 15 to 100 kDa (Figure 2).

### 2.3. Bioactivity Assay of F75AS and F90AS Fractions on F. Langsethiae Growth

Biological assay was performed to verify whether the different fractions (F75AS and F90AS, 80 μg each) had an inhibiting effect on *F. langsethiae* growth. The assay was performed on a Biolog FF Microplate with 96 wells, as reported in Materials and Methods. F75AS and F90AS fractions inhibited fungal growth at different level (Figure 3). F90AS fraction inhibited fungal growth both at 4 and 7 dai more than the F75AS fraction. In fact, the results showed that F90AS was more efficient than F75: 71.1% versus 11.1% at 4 dai (*t* test, *p* < 0.001 and *p* = 0.06, respectively) and 82.4 versus 46.4 at 7 dai, respectively (*t* test, *p* < 0.001 for both treatments).

The effect of both fractions on the fungal growth during the time was unchanged.

### 2.4. Further Fractionation of F75AS and F90AS by Sephacryl S-100

The protein fractions F75AS and F90AS were subsequently fractionated by size exclusion chromatography (SEC), using Sephacryl S-100 column (Appendix A). The chromatograms obtained from F75AS (Appendix A) and F90AS (Appendix A) allowed to individuate protein sub-fractions numbered as F75 (1–8) and F90 (1–6) based on their retention time and molecular weight (MW). As evidenced, in F75-SEC (Appendix A) different peaks were showed; this indicated that proteins were fractionated within a wide range of MW. The F90-SEC (Appendix A) shaped more like a Gaussian curve indicating a narrower MW range of the proteins.

### 2.5. Bioactivity of the Sub-Fractions F75 (1–8) and F90 (1–6) on F. Langsethiae Growth and Mycotoxin Production

The bioactivity of the sub-fractions obtained from F75AS and F90AS was analyzed in the same experimental conditions of the previous assays. After 7 dai at 25 °C the protein fractions F75_7 and F90_2, 4, 5 markedly inhibited fungal growth (Figure 4A,B). Concerning the F75 sub-fractions, up to 2 dai of no significant growth inhibition in respect of the control was evidenced. After this time the fractions F75_1, 2, 6, 7, 8 worked better than the others. As regards the F90 sub-fractions, the trend of growth inhibition is similar to the F75 sub-fractions up to 2 dai. After this point, in this case all the fractions, at different level, were able to inhibit the fungal growth. Considering the last time point at which the growth was registered, i.e., 7 dai, the fractions F75_7, F90_2, F90_4 and F90_5 were the most effective in limiting the fungal growth. Under these conditions, the T2 toxin was analyzed at 7 dai. At this time (7 dai), we tested the ability of the different sub-fractions in limiting T-2 toxin biosynthesis by *F. langsethiae* under the same cultural conditions (Figure 5). F90_4 and F90_5 inhibited about 60% and F75_7 and F90_2 about 98% T2 biosynthesis. Concerning the relationship between T2 production and fungal growth, as shown in the Table 1 below, the fungal growth resulted similar in F75_7, F90_2, 4, 5 fractions, in comparison with the control but the T2 toxin production was 2% in F75_7 and F90_2 and 40% in the other, respect to T2 production in the control.

### 2.6. Characterization of the Sub-Fractions F75_7, F90_2, F90_4 and F90_5

Considering the results about the fungal growth inhibition and T2 toxins biosynthesis, F75_7, F90_2, F90_4 and F90_5 fractions were further characterized by determining their isoelectric point (Appendix A as example) and proceeding with another type of purification. All the tested fractions presented a pI in the range of 4–5.5 (data not shown). The next purification step was done by using the anion exchange chromatography (Hi Trap Q). The proteins in the fractions were gradually eluted based on their different charge. In the Appendix A, we showed the chromatograms originated from the different fractions (Appendix A). All the fractions were pooled according to their affinity with the stationary phase and specifically divided into non-bound proteins, hereafter named as the “NL proteins” while the other (resin-bound) fractions were regarded as the “B proteins”.

Considering all the chromatograms (Appendix A) of the fractions of interest, they presented, in general, a widened and flattened trend typical of proteins distributed at different pH. As reported in Table 2, Bradford analysis showed that these fractions present lower protein quantity in comparison to F90_2 fraction. This latter was constituted by a high peak suggesting that this fraction was formed by a limited number of proteins with similar pI and present in high quantity (126.30 μg, Table 2). The sub-fractions obtained were collected and quantified by Bradford assay as reported in the Table 2.

### 2.7. Biological Assay of the Sub-Fractions

All the fractions obtained (as reported in Table 2) were assayed by Biolog FF microplate to assay their effect on growth and toxin inhibition in *F. langsethiae*. The tests were carried out in the same experimental conditions as the previous biological assays. As shown in Figure 6, every sub-fraction significantly inhibited the fungal growth. At 7 dai (Table 3), the untreated control produced 70.32 ppb of T2, whereas in the samples treated with different protein sub-fractions, the mycotoxin values were under the LOD (limit of detection: 0.01 ppb; normalized area 0.1) for every tested fraction. As evidenced, there was fungal growth inhibition and T2 toxin inhibition.

## 3. Discussion

*F. langsethiae* can be considered a recent entry in the *scenario* of plant pathogens, in fact it was described and accepted as new species and enclosed in a section within the genus *Fusarium*, *Sporotrichella* only in 2004 [15]. Unfortunately, besides its pathogenicity responsible for minor damage to small cereal grains and significant loss to their productivity, it also produces T2 and HT2 (a de-acetylated form of T2), mycotoxins belonging to the trichothecenes A type. Such toxins are responsible for altering RNA and DNA and causing lymphocyte depletion, ATA (Alimentary Toxic Aleukemia) in humans and animals. These considerations make the control of this pathogen, *F. langsethiae*, of paramount importance.

The European Union (EU) directives (e.g., 128/2009) concerning the methods to control fungal infection and mycotoxins in crops are now pushing research toward finding eco-compatible tools and integrated strategies to reduce the load of pesticides and fungicides burdening the environment. Further, this is even more relevant in the case of *F. langsethiae*, that is hardly controlled using fungicide spraying.

This study reports a novel tool to control the pathogen *F. langsethiae*—the partially purified exo-proteome of the nontoxic basidiomycete *T. versicolor*. Over recent years, the use of extracts from the carpophores of different basidiomycetes has received great attention mainly in medicine so that these mushrooms are named “medicinal mushrooms” for their therapeutic properties [2,3,4] and now they are considered also in medicine of developed countries. Here we propose the use of cultural filtrates of *T. versicolor* for producing and extracting bioactive compounds; in comparison with the mushrooms carpophores that require complicate procedures and techniques of cultivation, filtrates from mycelial culture are more easily to obtain. Preliminarily, we demonstrated that the total crude filtrate of *T. versicolor* cultures can significantly inhibit the growth of *F. langsethiae*. Our attention was then focalized on protein fractions of culture filtrates because previous studies, performed on other plant pathogens, evidenced the bioactivity of polysaccharide fractions on the inhibition of mycotoxins and not on the pathogen growth [6,9,11,12]. The rapid emergence of resistant microbial pathogens to currently available antibiotics has gained considerable attention from the scientific community in trying to isolate and investigate the mode of action of antimicrobial proteins (peptides) of natural origin (i.e., plants, fungi, bacteria) in consideration of the minor side effects in comparison with other drugs [21]. However, the majority of studies [1] concern peptides and/or proteins from fungi with antimicrobial and antibiotic properties applied mainly to human pathology. Some examples are plectasin, an antibiotic peptide from the saprophytic ascomycete *Pseudoplectania nigrella* [22] that demonstrated activity against *Streptococcus pneumoniae*, even against strains resistant to conventional antibiotics and RIP (ribosome inactivating proteins) from *Russula paludosa* that showed HIV-1 reverse transcriptase inhibitory activity [23]. However, some examples also regard the proteinaceous compounds from filamentous fungi applied in agriculture notably against plant pathogens—restrictocin, ribonuclease and ribosome inactivating protein from *Aspergillus fumigatus* and *A. restrictus* demonstrated suppressive activity toward *Fusarium oxysporum*, *Colletotrichum gloeosporioides*, *Trichoderma viride* and others [1].

Therefore, we focalized our attention on proteins constituting almost 15% of the cultural filtrate dry content of *T. versicolor*. To this aim, we carried out different steps of purification and characterization of the exo-proteome isolated from culture filtrates of *T. versicolor* looking for a protein fraction with inhibiting efficacy on the growth of the pathogen and toxin production in in vitro experiments. Ammonium sulfate precipitation at 75% and 90% provided different profiles—the former more uniformly distributed in a large range whilst the latter more focused within a higher MW range. In relation to their ability to inhibit fungal growth, the fraction F90 (F90AS and some of their sub-fractions) demonstrated particularly interesting, in view of a future subsequent purification with more sophisticated techniques. In particular, F90AS presenting several proteins with molecular weight ranging from 40 to 75 kDa and a pI ranging from 4 to 5.5. This fraction was able to inhibit about 90% the growth and T2 toxin production (98%) by *F. langsethiae*. However, not always fungal growth and mycotoxin synthesis are directly correlated as evidenced in the results obtained using the sub-fractions F90_2, F90_4 and F90_5, reported in Table 1. This phenomenon was already reported in previous studies on other pathogens and other toxins [24] where the authors underlined that the aflatoxin biosynthesis and the growth were not correlated in *A. parasiticus* and *A. flavus*. SEC-fractionation and bioassay indicate that F90_2 was the most efficient in inhibiting *F. langsethiae* growth as well as T2 toxin biosynthesis. Further, by the analysis with HiTrapQ, the sub-fraction B F90_2 (anion exchange resin-bound) results the most promising for application as antifungal. Firstly, it presents a high and clear peak that leads to suppose that this fraction is formed by a low number of proteins with similar pI in comparison with the other fractions that present a large range of pI; secondly, it is produced in high amount, respect to the other and lastly, completely inhibited fungal growth. However, at this purification step we still have fractions not completely purified and this makes impossible, at this time, to overexpress only one protein in heterologous systems.

In conclusion, this study concerns the application of bioactive compounds from *T. versicolor* to control the development and toxin production of the plant pathogen *F. langsethiae* and, at our knowledge, it is the first report concerning biological control of the recently discovered new species *F. langsethiae* that represents a serious problem for human and animal health for the synthesis of the dangerous T2 and HT2 toxins. The identification of the sub-fraction F90_2, probably formed by a low number of proteins, resulted promising for subsequent studies of purification and characterization towards the search of one protein inhibiting *F. langsethiae.*

Our previous studies and this report further confirm the *T. versicolor* 117 isolate as an effective source of bioactive compounds (polysaccharides and proteins) and as a promising tool in plant/fungal pathogens control and different mycotoxin inhibition.

## 4. Materials and Methods

### 4.1. Fungal Isolates

The biocontrol proteins were purified using culture filtrates of the non-toxic and edible basidiomycete *Trametes versicolor*, isolate117 from the fungal collection of the Department of Environmental Biology, University of Rome, Italy. *T. versicolor* 117 was kept in Potato Dextrose Agar (PDA, Difco, BD, Milan, Italy) a 4 °C for 30 days before the use. The inhibiting effect of *T. versicolor* exo-proteome was assayed on the plant pathogen *Fusarium langsethiae*, CREA-DC, Rome, Italy ER-1595 isolated in November 2009 from durum wheat (*Triticum turgidum* ssp. *durum*) and belonging to the fungal collection of CREA-DC. The fungal isolate was kept in PDA medium in test tubes under sterile mineral oil at 4 °C and incubated on PDA plates at 24 °C in presence of NUV (Near Ultra Violet) light before the use.

### 4.2. Growth Conditions and Rough Filtrate Production of T. Versicolor Cultures

The basidiomycete was grown in Petri dishes (9 cm diameter) for 7 days on PDA and incubated at 25 °C. After 7 days, 3 plugs (1 cm diameter) of agarized mycelial mass were added to 100 mL Potato Dextrose Broth (PDB, Difco, BD, Milan, Italy) in Erlenmeyer flasks and incubated at 25 °C for 14 days in shaken conditions (150 rpm). After the incubation period the total mycelia mass was homogenized in a Waring blender 8012 (Waring Conair, Stamford, CT, USA) and 5% *v*/*v* of the suspension was inoculated in 500 mL PDB and incubated at 25 °C for 14 days, in shaken conditions (150 rpm). After 14 days the cultures were filtrated by sequential filtration with 25 μm, 0.45 μm up to 0.2 μm filters Whatman to separate the mycelium from filtrates. Then the filtrates were concentrated 20 times with Rotavapor (Rotavapor^®^ R-300, Buchi, Essen city, Germany) and used for the experiments on *F. langsethiae* growth and toxin T2 and HT2 production.

### 4.3. Exo-Proteome Precipitation from Culture Filtrate of T. Versicolor

The culture filtrate was concentrated 25 times in Rotavapor as previously described and a protein precipitation in ammonium sulfate (AS) was carried out at 75% *w*/*v* concentration and kept in shaken condition at 4 °C over night (o.n.). Subsequently, the sample was centrifuged at 9000 rpm at 4 °C for 45 min. The pellet was recovered and resuspended in 20 mM KH_2_PO_4_, pH 7.2 and dialized (CO: 1000 Da) versus the same buffer. The supernatant was further precipitated with AS 90% *w*/*v* and kept o.n. at 4 °C. With the same previous procedure, the pellet was resuspended in 20 mM KH_2_PO_4_, pH 7.2 and dialyzed (CO: 1000 Da). The proteins were then quantified by Bradford assay (Bio-Rad, Hercules, California, USA) and lyophilized. The two fractions, F75 AS and F90 AS, were assayed for their bioactivity with the Biolog FF microplate analysis on *F. langsethiae* in vitro experiments.

### 4.4. Size Exclusion Chromatography, SEPHACRYL S-100

To further separate the F75AS and F90AS fractions based on their molecular weight, a size exclusion chromatography (SEC, Bio-rad, Hercules, CA, USA) with SEPHACRYL S-100 column HR 120 mL (GE Healthcare Life sciences, Chicago, Illinois, USA) was used. This column was designed for the separation of peptides and small proteins (i.e., fractionation range for globular proteins of 1 × 10^3^–1 × 10^5^). It was equilibrated first with 0.5 volume of H_2_O and subsequently with 2 volume of mobile phase (20 mM buffer KH_2_PO_4_ pH 7.2) at 1 mL/min flux. The samples lyophilized containing F75AS and F90AS proteins were suspended in 1 mL of buffer KH_2_PO_4_ pH 7.2 at 20 mM concentration and centrifuged at 12,000 rpm for 15 min to eliminate possible precipitated proteins. Then, the column was loaded with the 2 fractions and eluted with 20 mM of KH_2_PO_4_ buffer at pH 7.2 at 0.5 mL/min continuous flux. After eluting 40 mL, 1.5 mL sub-fractions were recovered and pooled considering the chromatogram obtained and quantified by Bradford assay and reported as F75 1–8 and F90 1–6. The fractions were dialyzed and lyophilized for the bioassays on *F. langsethiae*.

### 4.5. Protein Separation by pI—Isoelectric Focusing

To obtain a separation of the different proteins based on their pI with formation of bands with specific pH, we used polyacrilamide strips of Immobiline “Dry Strips” (IPG, Immobilized pH Gradient, BIO-RAD, Hercules, CA, USA), 7 cm long and applying a linear gradient from pH 4 to 7 and from 3 to 10. The focalization was carried out in a Mini-Protean III (BIO-RAD) apparatus. The protein fractions studied were F75-7 and F90-4 and, before charging, they were precipitated with methanol, chloroform and H_2_O to avoid the presence of buffer salts and to have proteins more concentrated. Subsequently, the fractions were suspended in about 2 mL of hydration buffer (2 M thiourea, 7 M urea, 2% CHAPS, (3-((3-cholamidopropyl) dimethylammonio)-1-propanesulfonate) 1% DTT (dithiothreitol), 1 mM PMSF (phenylmethylsulfonylfluoride), 0.1% protease-inhibitor cocktail) and shaken at 5000 rpm for 3 h for complete protein solubilization. Also, the strips were hydrated with a solution containing 9 M Urea, 2% CHAPS, 1% DTT, 2% di ampholytes pH 3.5–10 and 10 mM PMSF. The single fractions are charged on the strips and allocated in a tray for electrofocalization and after the addition of mineral oil the fraction were absorbed in the gel strip, passively for 2 h and actively for 12 h in presence of 50 V. Then, the samples were run for 1 day applying different voltages following a definite gradient—100 V for 1 h with rapid gradient, 350 V for 1 h with rapid gradient, 3 kV for 4 h with a linear gradient, 3 kV for 1 h with rapid gradient, 4 kV for 2 h with linear gradient and 4 kV up to achieve about 14 kV/h.

### 4.6. SDS PAGE

The protein separation on their molecular weight was carried out by SDS PAGE in polyacrilamide gel. The stacking gel was prepared by a final concentration of polyacrilamide (Biorad) 4% *w*/*v* in SDS 0.1% *w*/*v* in TRIS-HCl 0.5 M pH 6.8. The running gel was prepared by a final concentration of polyacrilamide 12% *w*/*v* in a buffer TRIS-HCl 1.5% pH 8.8 in SDS 0.1% *w*/*v*. To the samples of interest, a charging buffer (glycerol 10%, TRIS-HCl 0.06 M pH 6.8, bleu of phenol bromide 0.025%, SDS 2% and β-mercapto-ethanol 3%) was added. Before charging, the samples (F75% AS and F90% AS) were boiled for 5 min to favor the denaturation of the proteins. The electrophoresis was performed by MiniProtean III (Biorad) at 130 V. The protein fixing was carried out in a solution of MeOH 50%, CH3COOH 12% and Formaldehyde 0.02%. The silver staining was performed as reported by [25].

### 4.7. Ion-Exchange Chromatography—HiTrap Q HP Column

The sub-fractions most efficient in the inhibition of *F. langsethiae* growth, F75_7, F90_2, F90_4 and F90_5, obtained by SEC analysis, were further fractionated by ion-exchange chromatography. The use of the column HiTrap Q HP 5 mL (GE Healthcare Life sciences, Chicago, Illinois, USA) allowed the separation of small proteins and peptides by Q Sepharose High Performance as anionic exchange mean. The column was previously washed with 25 mL of start buffer (25 mM MES pH 6.05 + 20 mM NaCl) and 25 mL of elution buffer (25 mM MES pH 6.05 + 500 mM NaCl). The column was then equilibrated with start buffer for 10 min at 2 mL/min. The lyophilized samples were resuspended in 1 mL start buffer, centrifuged for 12 min at 12,000 rpm and loaded to the column. Then the column was washed with 25 mL of start buffer and the protein fraction not bound was collected as exclusion volume. The proteins in the samples were eluted with 10 volume of elution buffer. The elution was carried out by ionic gradient (0–100% elution buffer). The fractions eluted were collected in sub-fractions of 1 mL each and pooled after chromatogram and quantified by Bradford assay. At the end, the sub-fractions were dialyzed, stored at −80 °C and then lyophilized to be used in the biological assay.

### 4.8. Biological Assays—Biolog FF Microplate

The Biolog FF microplate (Hayward, CA, USA) was used to monitor the *F. langsethiae* growth in the 2 biological assays performed. Notably, in the first assay, the fungus was treated with the protein fractions F75 AS, F90 AS and the fractions obtained by SEC [F75_(1–8) e F90_(1–6)] at concentration of 80 μg in 190 μL in PDB; in the second assay, the fungus was treated with the sub-fractions obtained by the ionic-exchange column [F75_7, F90_2, F90_4, F90_5], at concentration of 60 μg in 190 μL of PDB. In both assays, the fungal growth was monitored. The Biolog system allowed to correlate the turbidity of the solutions of mycelium (treated and non-treated) with the fungal growth by different lectures at 750 nm. In both experiments, the microplate was constituted of 96 wells and the proteins were added together with a conidia suspension (100 conidia in 10 μL of sterile H_2_O) at a final volume of 200 μL for each well. The positive and negative controls were PDB + *F. langsethiae* and PDB alone. The plates were incubated at 25 °C for 0, 24 h, 48 h, 72 h, 96 h, 120 h e 168 h and analyzed by reading spectrophotometric absorbance at 750 nm.

### 4.9. Mycotoxin Extraction from Mycelia

T2 mycotoxin was extracted from all samples, treated and non-treated with proteins fractions, after 7 days of incubation. Mycotoxin extraction was performed by adding 1 volume of Acetonitrile/H20/Acetic acid 79:20:1 (*v*/*v*) to each sample. The mixture was shaker for 1 h, then centrifuged for 10 min at 13,000 rpm. The supernatant was collected and concentrated by drying in a stream of nitrogen. The samples were suspended in 200 μL of Acetonitrile/H20/Acetic acid 20:79:1 (*v*/*v*) and analyzed by in LC-MS/MS. The extraction was carried out on mycelia grown in culture media (PDB) and the internal standard U-[13C34]-Fumonisin B1 (Romer labs, Newark, DE, USA) deuterate was added at the final concentration of 0.5 μM, in order to normalize the chromatogram area of the different samples.

### 4.10. Quantification of T2 Toxin by LC-MS/MS

The quantitative analyses were performed by a LC-MS/MS (Agilent Technologies, Santa Clara, CA, USA) equipped by an LC (HPLC 1220, rapid resolution) coupled with a triple quadrupole MS (G6420 triple quadrupole, QQQ, Agilent Technologies) equipped with an electrospray ionization source (ESI). The chromatographic separation was carried out by Poroshell 120 EC-C18, 2.1 × 100 mm, 2.7 μm (p/n 695775-902) column, Agilent Technologies (USA). The elution phases were—mobile phase A: H2O/MeOH/Acetic acid 89:10:1 *v*/*v* and eluent phase B: H_2_O/MeOH/Acetic acid 2:97:1 *v*/*v*. Both phases presented 5 mM ammonium acetate. The elution program is reported in Table 4.

The column was heated at 60 °C and the flux fixed at 0.6 mL/min. The injection volume was 10 μL for each sample analyzed. Nitrogen was used as nebulization and desolvation gas. For the quantitative analysis was applied the Multiple Reaction Monitoring (MRM) method to detect simultaneously several toxins. The toxins and the parameters utilized for their determination are reported in the Table 5 [26].

### 4.11. Statistics

Error bars represent the standard error calculated from three independent replicates, technically repeated in trice. XLSTAT and Rstudio were used as the statistics packages. Datasets were pooled and compared using one- or two-way ANOVA followed by Tukey test and the differences were considered significant when the *p*-value was <0.05 or in other cases <0.001.

## Figures and Tables

**Figure 1 ijms-20-04167-f001:**
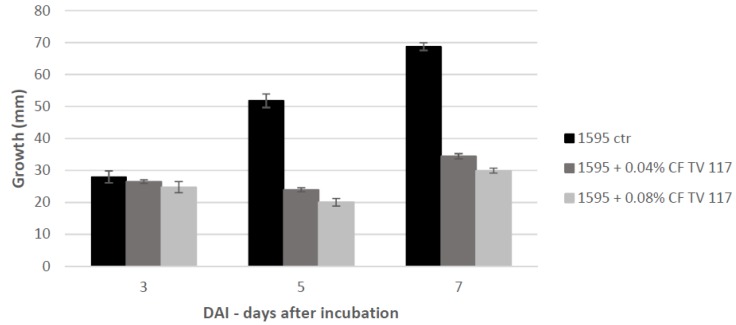
Growth inhibition of *F. langsethiae* (1595) treated with 0.04 and 0.08% *w*/*v* of CF TV117. Histograms of the growth during the time (3, 5 and 7 dai) and the percentage of the growth inhibition in comparison with the control (ctr) (100%) is reported. The data are the mean ± SD of 3 experiments.

**Figure 2 ijms-20-04167-f002:**
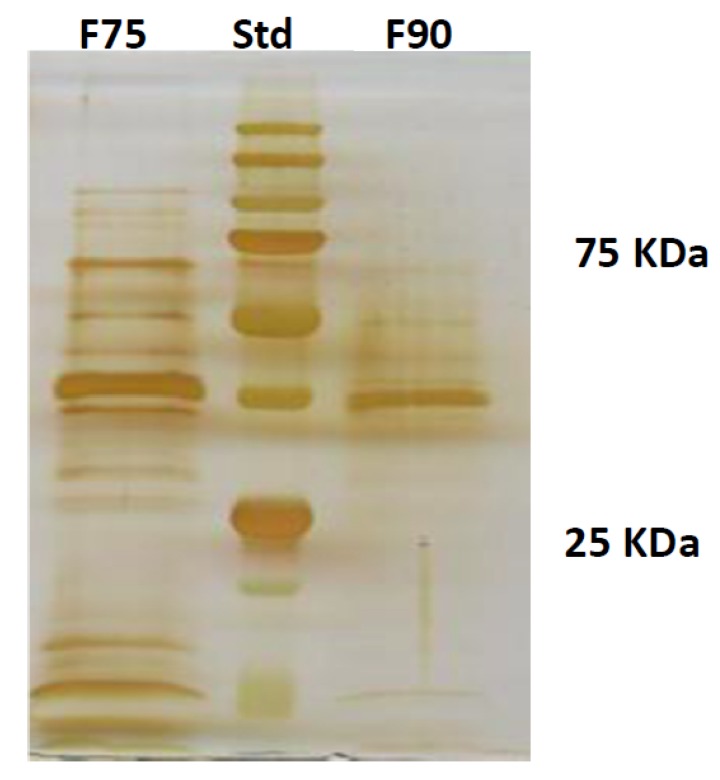
SDS-PAGE of F75AS (left line) and F90AS (right line) protein fractions. Std is the protein marker (middle line). The molecular range of std is from 11 to 245 kDa.

**Figure 3 ijms-20-04167-f003:**
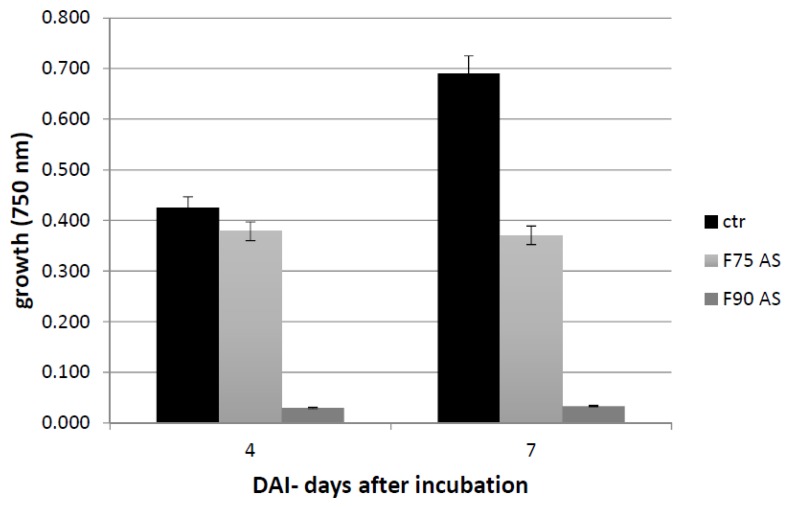
Fungal growth (absorbance at 750 nm) assayed by Biolog FF microplate, after 4 and 7 dai at 25 °C in presence (F75 AS and F90 AS) and in the absence (ctr) of protein fractions. The results are the mean ± SD of 3 different experiments.

**Figure 4 ijms-20-04167-f004:**
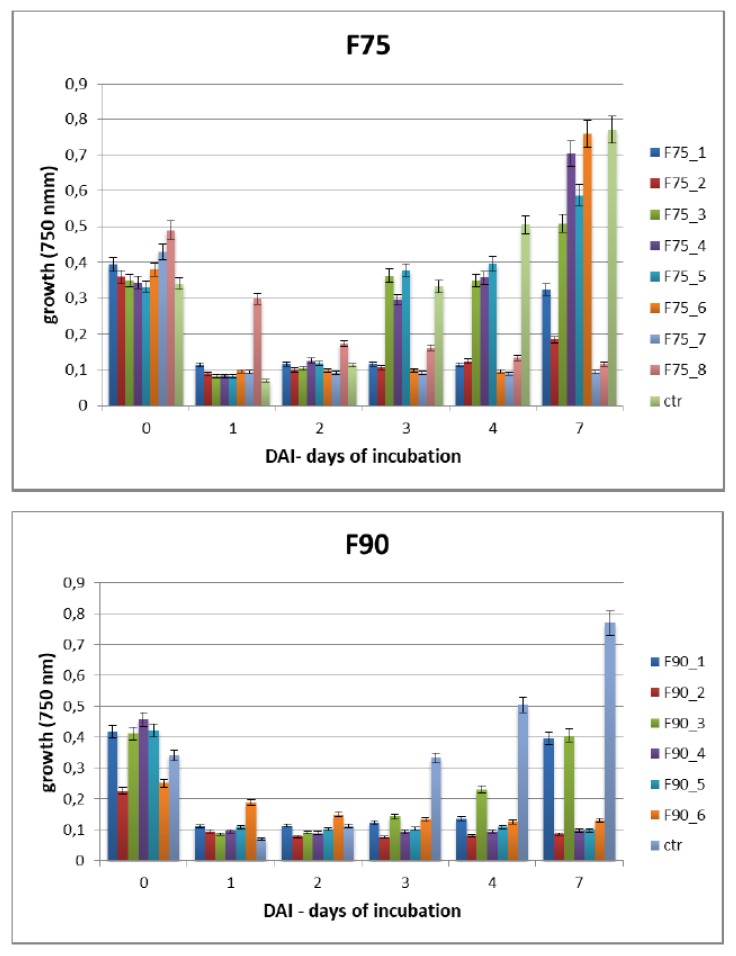
*F. langsethiae* growth analyzed by Biolog FF microplate analysis at 1, 2, 3, 4 and 7 dai at 25 °C. Untreated control (ctr) and samples treated with F75 1-8 (**A**) and F90 1-6 (**B**). The results are the mean ± SD of 3 different experiments.

**Figure 5 ijms-20-04167-f005:**
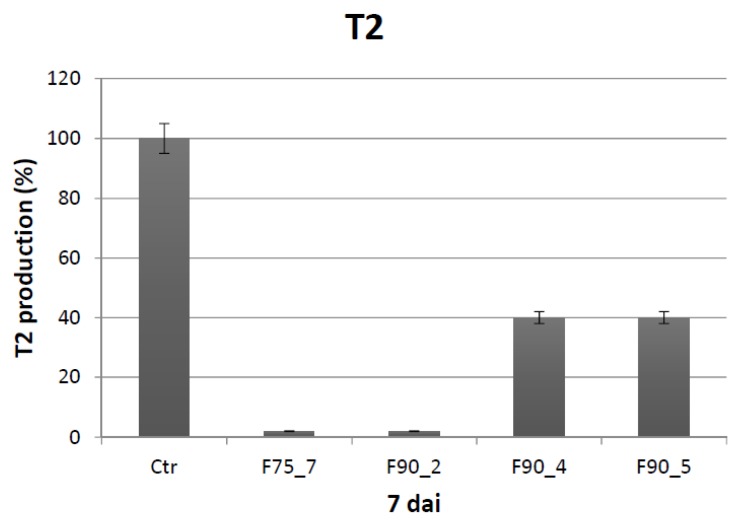
Percentage (%) of T2 toxin produced from *F. langsethiae* after 7 dai at 25 °C in the absence (Ctr) and presence of sub fractions F75_7, F90_2, F90_4, F90_5. The results are the mean ± SD of 3 different experiments.

**Figure 6 ijms-20-04167-f006:**
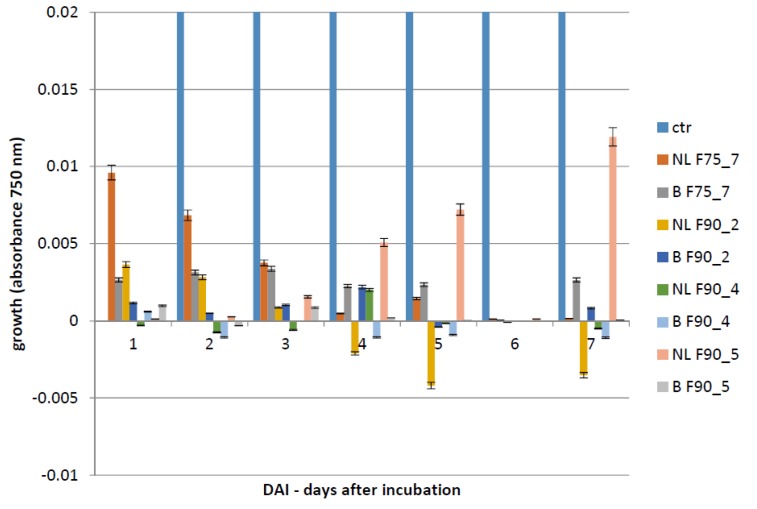
Biological assay of the *F. langsethiae* growth in samples non treated (ctr) and treated with the fractions indicated in the graph legend after 1, 2, 3, 4, 5, 6, 7 dai. The results are the mean ± SD of 3 different experiments.

**Table 1 ijms-20-04167-t001:** Fungal growth and T2 toxin production by *F. langsethiae* non treated (ctr) and treated with protein fractions F75_7, F90_2, F90_4, F90_5 at 7 dai at 25 °C.

Fractions	Growth (%)	T2 Toxin (%)
ctr	100	100
F75_7	12.5	2
F90_2	11.7	2
F90_4	12.5	40
F90_5	12.5	40

**Table 2 ijms-20-04167-t002:** Bradford analysis of the fractions (bound, B and non-bound, NL) F75_7, F90_2, F90_4, F90_5.

Fractions	Proteins (µg)
NL F75_7	12.50
B F75_7	56.40
NL F90_2	36.50
B F90_2	126.30
NL F90_4	121.25
B F90_4	37.30
NL F90_5	74.23
B F90_5	64.80

**Table 3 ijms-20-04167-t003:** T2 production by *F. langsethiae* treated and non-treated (ctr) with the different fractions after 7 dai at 25 °C.

Fractions	T2 Toxin (ppb)
Ctr	70.32
NL F75_7	<LOD
B F75_7	<LOD
NL F90_2	<LOD
B F90_2	<LOD
NL F90_4	<LOD
B F90_4	<LOD
NL F90_5	<LOD
B F90_5	<LOD

**Table 4 ijms-20-04167-t004:** Elution program.

Time (min)	A%	B%
0	80	20
6	80	20
14	2	98
18	2	98
20	0	100
22	0	100
26	80	20
28	80	20

**Table 5 ijms-20-04167-t005:** Multiple Reaction Monitoring (MRM) parameters for the mycotoxins quantifications.

	Precursor Ion	Product Ion	RT (min)	CE (V)	Fragmentor (V)
**T2**	484.3	185	14.746	12	20
**HT2**	442.2	263.1	11.222	9	20
**15ADON**	339.0	261.0	8.850	10	20
**3ADON**	339.0	203.0	8.852	10	20
**FB1 [13C34]**	756.4	374.4	11.612	37	135

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
