# Peer review of "Use of the Secreted Proteome of Trametes versicolor for Controlling the Cereal Pathogen Fusarium langsethiae"

_ijms, 2019, doi:10.3390/ijms20174167_

Round 1

Reviewer 1 Report

The topic of finding new microbial inhibitors from natural resources is both interesting and timely as we face the risk of development of resistance to common fungicides. Also, F. langsethiae is a toxin producing FHB pathogen that is hardly controlled by the same conditions for fungicide spraying as F. graminearum, making it more imperative to find new control measures.

In my opinion this kind of research should be encouraged, however, the manuscript in the present state needs thorough revision which I have tried to outline below.

Three main issues: In my opinion the results are maybe not as clear as the discussion gives the impression of, and a genuine critical discussion is warranted. Also, additional experiments would be an enrichment to the paper and something to consider: e.g.  test several F. langsethiae strains, compare response in F. langsethiae to other Fusarium species, test inhibition in planta (e.g. in a detached leaf assay), define growth inhibition by also checking inhibition of spore germination, supply with other separation methods. The inhibition of T2 biosynthesis needs to be normalized to fungal mass in order to show inhibition independent of fungal growth.

Introduction: The rational for choosing extracts from Trametes versicolor is well described, however, the introduction could be structured better and also include more rational for choosing F. langsethiae among all the plant pathogens. Some copyediting is needed to improve language: prepositions, plural or singular, “a” is missing, past/ present tense.

Results: The results section is challenging to follow and it is hard for the reader to judge from some of the figures if the conclusions are reasonable. Restructure the sections and maybe put some together, it needs to be made more clear what exactly was done and names/abbreviations used in the material and methods should also be used consistently in the results (e.g. F75AS and F90AS). The interpretation and meaning of the results needs to be explained in more detail. Figures can be made more self-explainable by fixing the labels, figure 4B is missing, and figure 6 needs to be in color. Figure 6 is also very hard to conclude anything from the way it is presented as the control determines the y axis. I also suggest to put a table summarizing the results of all the fractions with effect on growth and T2 biosynthesis as well as data on protein size, pI, and ion-exchange bound/non-bound. I wonder why the different bound fractions from the ion-exchange were pooled? I would also like to see a normalization of T2 toxin to fungal biomass to determine whether T2 production is indeed reduced or simply a result of growth inhibition. The authors comment on this in the text, but it is not clear to the reader what they base the conclusion on. Additionally, could growth inhibition and T2 production be integrated into the same figure?

Discussion: Some restructuring of the text is needed. As I see it parts of the beginning could go to the introduction or it is actually a repetition of the introduction. Furthermore, the conclusion gives the impression of being a little hasty and is not firmly based on what is described in the results. Alternatively the results need to be better described (see above). I miss more discussion about the results originating from F90AS as the authors also remark that this fraction seems to have poor protein quality. Could the results be due to artefacts? Would other separation methods be needed? This needs to be discussed. Also, the inhibition of T2 production needs to be discussed in more detail as to whether or not it is a result of reduced fungal growth. Maybe additional experiments need to be performed? I miss a critical discussion of the impact of the results with respect to implications of use as well as to the fact that the fractionation methods chosen did not succeed to separate out “clean” fractions.

Concrete remarks:

Page 2 results and other places: culture filtrate should be CF, not FC Page 3 and other places: indicate F75 and F90 with AS where appropriate as it is written in material and methods. Page 3 line 19: F75AS and F90AS are not a result of molecular exclusion chromatography Page 4 line 15: lacking reference to figure? Page 4 line 24-26: needs more elaboration and explanation, pluss normalization of T2 to fungal biomass. Figure 4B is missing. In figure legends: correct accordingly to be consistent about number of independent replicates and technical reps. Page 5 line 10-12: not clear that only 2 fractions were tested with isoelectric focusing. Why not all? Page 6 line 3: change “bound proteins” to B proteins. Be consistent about hrs or days of incubation. Page 8 line 24: isolate is not CF117, change to TV117? Editing remarks still present in the text in supplementary Correct “(dai?)” in figure legend figure 5. Why is table 4 showing fum B1 and DON? Page 10 line 24: internal standard is fumonisin B1?

Author Response

Thank you for the careful revision of MS and suggestions.

The English was revised.

- Introduction:

The section introduction was improved as suggested by the referee and rationale for the choice of F. langsethiae is proposed.

- Results:

 The section results as revised as the referee indicates. Each paragraph of results is more detailed as requested in this manuscript we believe that the revision of figure legends make them itself more comprehensible and in line with the discussion section. The figure 4A is now embedded in the text at the right place. The figure 6 is now in colour and the scale changed to make the results obtained more easy to read and to appreciate the differences among the samples.

A table that correlate fungal growth and T2 toxin production is added, as requested by referee. A way to correlate fungal biomass and T2 quantity is presented.

- Discussion:

Concerning the section discussion the suggestion posed by referee are followed. We improve the discussion about the protein fraction F90 through better explanations.

We better discuss the correlation between T2 reduction and fungal growth. A more detailed and critical discussion is added, as requested, about the use of fractions not pure.     

The suggestions of the referee about other different experiments concerning different isolates of F. langsethiae and others are very interesting for future research on this pathogen. Unfortunately, the little time available makes impossible to carried out the experiments that referee suggested.

Reviewer 2 Report

The results presented in the manuscript show an inhibiting effect both on mycelial growth as well as on the production of certain mycotoxins in a strain of F. langsethiae when cultures are confronted with fractions of a culture filtrate of a strain of T. versicolor that have been manipulated using techniques  typically applied in protein chemistry. However, this fact does not proof that proteins are the source of the observed effects, which might also have been induced by non-proteinaceous compounds present in the extracts. The classic way to prove that the effective inhibiting principle is a protein would be when the effect can be abolished upon protease treatment of the tested fraction. Moreover, since authors discuss their results in regard to their future application in the field, it might be useful to demonstrate that the observed inhibition of growth and mycotoxin production is a more general phenomenon. This would be done by analyzing not just one but a few more F. langsethiae strain that were isolated from different hosts and/or from different regions. The suggested additional data would make the results much more convincing.

Overall, the text is written in a clear and concise manner. The background that has led to the study, the aims of the study as well as the roles of donor and acceptor organisms have been comprehensively described. The results are clearly described and documented with figures and tables. The discussion part might need some modification in regard to the few experiments I have suggested to be performed in the previous paragraph. Much of the text does not fulfill the requirements of Standard English. Author struggle with tenses as well as with the correct endings of plurals and singulars. Moreover, sentences are often quite long and should shortened. In many cases, this can be achieved by just entering a full stop instead of a comma. In addition, authors often use a semicolon where a full stop would be the better choice. Editing of the text by a native speaker or a person with lots of experience in scientific writing in English is highly recommended.

Specific comments

Page 1, line 43: …. and this family is more and more increasing”. This phrase is incomprehensive since it doesn´t tell to what this increase relates.

Page 2, line 1: mycotoxins instead of toxins.

Page 2, line 3: I am not sure whether trichothecenes really have a cancerogenic effect on the nervous system, as stated here.

Page 2, line 4: “…. Non-toxic or edible basidiomycetes such as Lentinula edodes and T. versicolor.” This implicates that T. versicolor is an edible mushroom. I guess authors want to say that it is not dangerous to eat T. versicolor. An edible mushroom is something that would be used in a meal, which T. versicolor will never be, except as table decoration due to its impressive ornamentation.

Page 2, line 40: “….. culture filtrate (FC)”. Make sure in the entire manuscript to use CF instead of FC

Page 2, line 43: dai stands for “days after inoculation” and not for “days of incubation”. Please correct in entire manuscript and in figures.

Page 3, line 19: size exclusion instead of molecular exclusion

Page 4, line 19: …. (Figure 4 A and 4 B). Figure 4 is supposed to show data about fraction F75 1-8 and F90 1-6. However, the figure does only contain information about fractions F90 1-6. Please correct!

Page 6, lines 18, 19: This last sentence of the results section should be moved to the discussion part and the question should be discussed there.

Page 7, line 4: Please give the original citation (Torp and Nirenberg, 2004) for the description of F. langsethiae. Sporotrichiella is a section within the genus Fusarium, not family.

Page 8, line 28: add “(Rome, Italy)” to CREA-DC.

Page 8, line 44 and elsewhere: Please do not use rpm as a unit for centrifugation because the centrifugation capacity depends on the rotor diameter used. Better, use rcf-values with “x g” as unit for the re-calculated value. Please check in the entire text.

Page 8, line 45: mM KPi instead of mMKPi. Even better to use KH2PO4.

Page 9, line 15: isoelectric focusing instead of isoelectrofocusing

Page 9, line 47 and elsewhere: … and charged in the column. Better to use loaded to the column. Please check in entire document and use load instead of charge.

Figure S3: not much to see on these gels. Better, add some arrows to mark the items of interest on this figure.

Author Response

Thanks for your kind suggestions. We trying following them by closely.

We agree with the referee that the exo-proteome could have some impurity and in future we will test fractions treated with protease and we intend to verify the composition by more sophisticated techniques.

However, in previous study (Scarpari et. al., 2017) to obtain the pure polysaccharide Tramesan from CF of T. versicolor, pronase was used and after this treatment the growth of Aspergillus flavus was not inhibited in comparison with the control.

We follow, page by page, the corrections requested by the referee.

About the denomination “edible” of T. versicolor, we believe that this can be used since in Oriental countries this basidiomycete is used in soups and beverages as teas.   

Round 2

Reviewer 2 Report

General comments

Revision of the manuscript greatly improved the text for readability and mistakes. However, I have still some concern regarding the consistent use of past tense in the description of the results and in cases where the discussion relates to results obtained during the current study. In addition, the text would become more readable by shortening some very long sentences or by making few shorter sentences from one lone sentence. Authors should avoid the use of semicolons in a sentence and better replace by a full stop. Another concern is the use of “dai” which now translates into “days after incubation” according to the authors. However, this is still incorrect and I have commented on this in the specific comments to the authors. Typical filling words and pharases such as “in fact” should be avoided in a scientific text.

Specific comments

Page 1, line 11: renders instead of posit

Page 1, line 34: delete “in fact” after Tramesan

Page1, line 34: …. Expression and protected melanocytes ….

Page 1, line 36: … cell growth was observed.

Page 1, line 41: …. molecular weight around ….

Page 2, line 1: …. agriculture. Indeed, mycotoxins are ….

Page 2, line 4: … Polysaccharide fractions ….

Page 2, line 10: …. Strengthening the defense of plants against ….

Page 2, line 16: Previous research evidenced …

Page 2, line 21: …. barley). The fungus is very detrimental for crop quality ….

Page 2, line 22: … of different toxins, some of which ….

Page 2, line 28: … in Central and Southern ….

Page 2, line 31: … their lipophilicity making them likely to penetrate …

Page 2, line 35: … small grain cereals …

Page 2, line 37: …. safety of and security of …

Page 2, line 38: …. about the prevention of F. langsethiae. …

Page 2, line 47: comment: days after incubation makes no sense here. Please use days after inoculation here and in all figures or switch to “days of incubation” or “incubation time”. Days after incubation would mean that cultures are first incubated and that the analysis is done after a certain number of days after the incubation has stopped. I am sure this is not what the authors had in mind.

Page 2, line 47: … At 5 dai the colony diameter decreased by 53.8 % and 61.4 % compared to the untreated control when 0.04 % (w/v) and 0.08 % (w/v) of culture filtrate was added, respectively.

Page 3, line 12: comment: what does “a lower type of protein” mean? Is it lower molecular weight? Is it a lower intensity of the protein bands in the gel? Please specify!

General comment for Results: use past tense consequently to describe the results of your experiments, e.g. Page 3, line 14 … fractions assayed were in the range … instead of … fractions assayed are in the range….

Page 3, caption for fig. 2: please replace the word “line” by the word “lane” and replace “std” by “Std”

Page 4, line 17: … different peaks were observed ….

Page 4, line 22: …. no significant growth inhibition was observed as compared to the control.

Page 6, line 8: Considering the results of fungal ….

Page 6, line 15: better use NB for non-bound and B for bound proteins?

Page 8, line 3: …. plant pathogens. In fact it was …

Page 8, line 4: … enclosed in section Sporotrichiella of the genus Fusarium only in ….

Page 8, line 13: … which is hardly …

Page 8, line 19: …. bioactive compounds. In comparison ….

Page 8, line 22: Our attention subsequently focused on …

Page 8, line 37: … we focused our …